

# Advancement in the development of single chain antibodies using phage display technology

Xiaohui Zheng, Qi Liu, Yimin Liang, Wenzhi Feng, Honghao Yu, Chunyu Tong and Bocui Song

College of Life Science and Technology, Heilongjiang Bayi Agricultural University, Daqing, Heilongjiang, China

## ABSTRACT

Phage display technology has become an important research tool in biological research, fundamentally changing the traditional monoclonal antibody preparation process, and has been widely used in the establishment of antigen-antibody libraries, drug design, vaccine research, pathogen detection, gene therapy, antigenic epitope research, and cellular signal transduction research.The phage display is a powerful platform for technology development. Using phage display technology, single chain fragment variable (scFv) can be screened, replacing the disadvantage of the large size of traditional antibodies. Phage display single chain antibody libraries have significant biological implications. Here we describe the types of antibodies, including chimeric antibodies, bispecific antibodies, and scFvs. In addition, we describe the phage display system, phage display single chain antibody libraries, screening of specific antibodies by phage libraries and the application of phage libraries.

## INTRODUCTION

Since 1975, Kohler (*Köhler & Milstein, 1975*) has generated specific monoclonal antibodies (mAbs) against antigens by fusing mouse bone marrow cells with mouse spleen cells, resulting in the production of hybridoma cells. MAbs were highly homogeneous in structure and composition and can be easily prepared and purified in large quantities *in vitro*, such that they have been extensively employed in clinical diseases and to detect specific antigens (*Schönherr & Houwink, 1984*). MAbs can be used for sclerosis (*Krajnc et al., 2022*), respiratory diseases (*Boonyaratanakornkit, Boeckh & Waghmare, 2022*) and immune thrombocytopenic purpura (ITP) (*Gómez-Almaguer, 2012*). There were several methods to produce mAbs, including phage display technology, hybridoma technology, single B cell culture, immortalized B cells, and B cell sorting (*Merkuleva, Shcherbakov & Ilyichev, 2022*). Although mAbs exhibited high purity, specificity, and potency, there were limitations to the advancement of mAbs. The preparation process of monoclonal antibody was cumbersome and the preparation technology was higher. The time and raw material costs were higher. The high technical requirements of the preparation were mainly

Corresponding authors
Chunyu Tong,
tongchunyu@126.com
Bocui Song, songbocui66@163.com

reflected in that the mouse bone marrow cells passaging process should keep the cells in the logarithmic growth period, otherwise the fusion efficiency will be reduced. When mouse spleen cells were fused with mouse bone marrow cells, the fusion ratio of the two was 10:1 to 5:2 (*Jahn et al., 1988*). In addition, the health of the cells and mice was very important. The technical requirements for mouse bone marrow cell culture were high. During the cell culture process, mycoplasma contamination was common, and the contamination of bone marrow cells will affect the expansion of hybridoma cells. Also, there was potential viral contamination in mice, such as mouse hepatitis virus, which activated the body's immunomodulatory response and affected antibody production (*Du et al., 2021*). Therefore, it was important to require healthy mice and uncontaminated cells. With the continuous progress of new technologies, genetically engineered antibodies continue to emerge. There was a wide range of genetically engineered antibodies, including bispecific antibodies, chimeric antibodies, scFvs.

In 1983, Aalberse initially observed and confirmed the natural bispecificity of antibody molecules. Bispecific antibodies can bind two different antigenic epitopes, respectively (*Aalberse, van der Gaag & van Leeuwen, 1983*). He introduced two sets of heavy chain and light chain genes into multiple myeloma cells. The selected appropriate antibody constant region and Ig type can get the large yield, homogeneity and high purity of bispecific antibodies (*Brinkmann & Kontermann, 2017*). Bispecific antibodies took on a significance to the treatment of tumors, with one antigen specific to the target antigen and the other antigen specific to the cell, directing the drug to the target cell and thus achieving treatment of the tumor (*Li, Er Saw & Song, 2020*). *Blanco, Domínguez-Alonso & Alvarez-Vallina (2021)* conducted immunotherapy with bispecific antibodies, which had greatly changed the treatment strategy for advanced malignant tumors based on the latest progress in the field of immunooncology. It had the potential to improve clinical efficacy and safety, and was envisaged as cancer immunotherapy. The development of cancer primarily involved Akt, MAPK/Erk and MTOR signaling pathway due to the complexity of cancer cells. Thus, the most effective therapeutic option refered to stopping the trajectory of cancer cells by targeting the above-mentioned signaling pathways simultaneously. Thus, bispecific antibodies were appropriate and optimal therapeutic agents for the treatment of such cancers (*Acheampong, 2019*). Many preclinical and clinical trials were currently underway, painting a bright future for cancer therapy with bispecific antibodies. Nevertheless, bispecific antibody drugs still face great challenges in their application as cancer therapeutics, including tumor heterogeneity and mutation burden, refractory tumor microenvironment (TME), insufficient co-stimulatory signals to activate T cells, the necessity of continuous injections, lethal systemic side effects, and off-target toxicity to adjacent normal cells.

Antibodies can be derived from animals, however, obtaining antibodies from humans poses ethical and moral challenges, preventing their acquisition. In 1984, *Boulianne, Hozumi & Shulman (1984)* published chimeric antibodies from mice and humans, constructing immunoglobulin genes. They produced human antibodies by ligating a DNA fragment encoding the variable region of the mouse specific synergist trinitrophenyl (TNP) to a fragment encoding the human μ and k constant regions. *Morrison et al. (1984)* ligated

variable region genes cloned from rats or mouse hybrid genes to human constant region genes, and the above-mentioned chimeric genes were transfected into human bone marrow cells, such that essentially human antibodies were produced. Due to it reduced mouse-derived components, it reduced the adverse reactions caused by mouse-derived antibodies and contributed to the efficacy of the treatment. It offered a new way to understand the structure, function and immune properties of antibody molecules. The use of chimeric mouse-human antibody gene structures allowed us to study antibodies with the required antigen binding specificity. In clinical applications, human antibodies established using recombinant DNA technology can serve as immunotherapeutic and diagnostic reagents (*Shin, 1991*). Chimeric antibody technology had wide promising applications (*e.g.*, in the purification of cloned gene products, enzyme-linked immunoassays, and serving as a vehicle for the targeted action of chemical drugs on target cells). Although the chimeric antibody weakened the human anti-mouse antibody response, there was still a small mouse-derived component. This led directly to the rapid removal of antibodies, which reduced the therapeutic effect.

Human antibodies were obtained by the technical means based on phage display single chain antibody libraries. Mammals, rabbits, camels and chickens have been part of the most commonly used host animals to produce scFvs. They produced high specificity and high affinity scFv (*Hudson, 1998*). ScFvs were the more commonly used antibodies in genetic engineering. ScFvs consist of the heavy chain variable region of antibodies (VH) and the light chain variable region of antibodies (VL) through a segment of flexible Linker in the middle to form VH-Linker-VL or VL-Linker-VH (*Ahmad et al., 2012*). Primers were designed with the constant regions FR1 and FR4 of the antibodies to amplify the VH and VL. VH and VL were linked by a section of linker $(GGGGS)_3$, usually between 10 and 15 amino acids, which should not be extremely short or excessively long. An extremely long linker sequence may reduce the yield of the fusion protein while leading to immunogenicity problems. If the linker sequence was excessively short, the two proteins may be too closely spaced, affecting the folding of each other's higher structures. As a result, the two proteins can interfere with each other, leading to protein loss of function (*Chen, Zaro & Shen, 2013*). Subsequently, scFvs was ligated to phage vectors like pcantab5e, and then after digestion, ligation, transformation, and helper phage infection, the phage display single chain antibody libraries were obtained (The process of constructing phage display single chain antibody libraries was illustrated in Fig. 1). The greatest advantage of the phage display single chain antibody libraries was that the variable regions of the antibodies comprise genes from highly variable regions, ensuring the diverse and large antibody libraries. Integrity of primers for the amplification of antibody libraries and transformation efficacy have been confirmed as a key reason for the capacity of the antibody libraries (*Xia et al., 2006*). *Kato & Hanyu (2013)* expanded the library capacity of the antibody libraries by enzymatic assembly. They established scFv by enzymatic assembly of VL and VH structural domains using synergistic action of λ-exonuclease and Bst DNA polymerase. In contrast, *Yang et al. (2009)* used a β-lactamase selection strategy to enlarge the functional variety of the libraries. They fused the installation of scFv genes sequence to the 5′ end of the β-lactamase genes and performed scFv cloning by penicillin
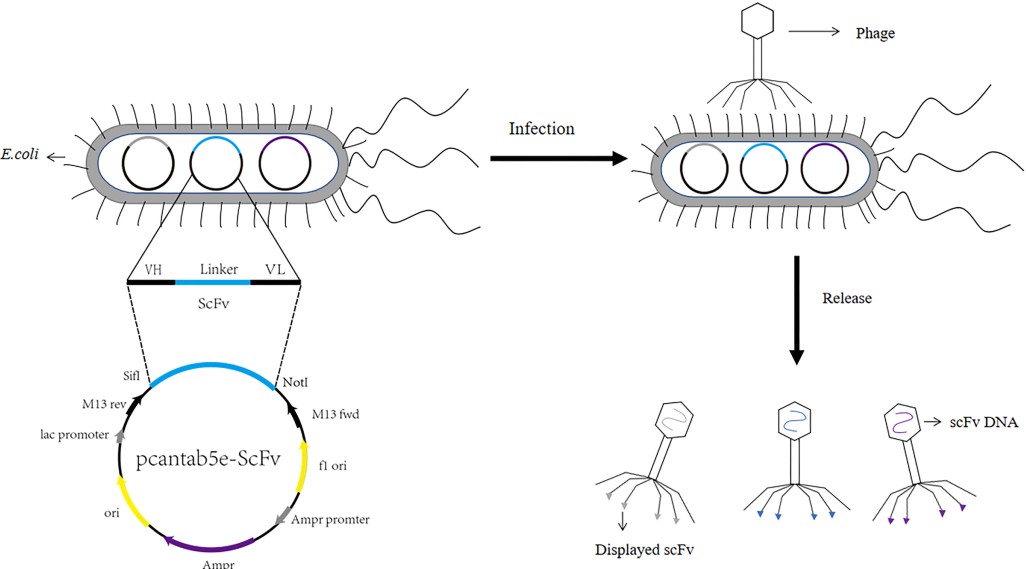

**Figure 1 Construction process of phage single chain antibody libraries.** The scFv is ligated to the phage vector (eg. pcantab5e), transferred into *Escherichia coli*, and eventually displayed on the surface of the phage using phage infestation.

choice. *Sblattero & Bradbury (1998)* adopted the Cre recombinase method, which the vector comprises two non-homologous Loxp sites, to build the large capacity human derived natural antibody libraries. The Cre-Loxp system was adopted to amplify the variety of the library.

In 1990, *McCafferty et al. (1990)* constructed an antibody library with a library capacity of 106 and successfully screened lysozyme scFvs from it. Phage display technology became an emerging means of antibody preparation in antibody engineering (*McCafferty et al., 1990*). The scFv gene was displayed on the surface of the phage through phage infection, which was screened from the phage display single chain antibody libraries using specific antigens. The final antigen enrichment was achieved after several rounds of screening (*Baca et al., 1997*). ELISA assayed the binding activity of the screened phage with the target molecule, and scFv was obtained after DNA sequencing. Currently, phage display single chain antibody libraries technology had an indispensable role in vaccine development and tumor therapy. We described the advantages of screening scFv using phage display technology and summarized the differences between T4, T7, λ and M13 phages. We also discussed the use of solid phase and liquid phase screening for scFv and phage antibody libraries applications.

## SURVEY METHODOLOGY

To search for articles related to advancement in the development of single chain antibodies using phage display technology. We searched the keywords "phage display, phage display antibody, phage display technology, monoclonal antibodies, bispecific antibodies, chimeric antibodies, scFv, phage display single chain antibody libraries, natural antibody libraries, semi-synthetic antibody libraries, synthetic antibody libraries, screening of phage display

single chain antibody libraries, phage libraries, T4 phage, T7 phage, λ phage, M13 phage" in PubMed. A total of more than 3,000 articles were retrieved. The search results were screened one by one as needed. Based on the keywords and other information to initially determine whether it meets our research needs. The screened literature was organized and analyzed. A total of 160 literature reviews were downloaded. A total of 105 articles were used for citation in the writing of this article, and 55 articles were excluded because they did not fit the topic of this article. Literature with publication date in the last decade was more. The purpose of this review was to provide a reference for researchers engaged in phage research and the construction of single chain antibody libraries using phage display technology.

## Phage display technology

The discovery of phages has aroused the attention of scientists at the beginning of the last century, and initially phages were useful for infecting specific bacteria or viruses. As technology developed, phages were thought to carry some types of genetic material and became an important part of the field of genetics (*Bertani & Weigle, 1953*). In 1985, *Smith (1985)* inserted foreign genes into phages, displayed the foreign genes on phage coat proteins, and used phage display technology to construct single chain antibody libraries that could be screened for sequences expressing a specific scFv. Phages were stable under external factors (*e.g.*, pH, temperature). Phage coat proteins can be modified to display foreign genes as fusion proteins without losing the infectivity of the phage. With the progress of technology, phage display technology has been matured, and phage peptide libraries, phage single chain antibody libraries and phage synthesis libraries can be established by phage display technology (*Sunderland, Yang & Mao, 2017*). Phage display can be used for the analysis of protein interactions as well as the screening of proteins or the selection of antibodies for diagnostic and therapeutic purposes, mimicking natural antibody libraries through billions of phage particles presenting different antibodies. The origin of the phage antibody libraries significantly affected on the species of phage antibody libraries. In general, the more diverse the sources, the greater the capacity of the phage antibody libraries (*Frenzel, Schirrmann & Hust, 2016*; *Beerli & Rader, 2010*). The main benefit of using phage display to obtain antibodies was that it was fast and did not need immunization of animals and humans. However, phage display systems included the T4, T7, λ and M13 phage display systems (The differences between them were shown in Table 1). Among them, M13 phage was the most common phage in the construction of phage display single chain antibody libraries.

## T4 phage

T4 phage was a virulent phage belonging to the T lineage of *Escherichia coli*. The head was icosahedrally symmetric and the tail was spiral symmetric, called tadpole-shaped phage. T4 phage was a cleavable phage that could replicate and proliferate in the host bacteria, producing many zygote phages, and eventually lysed the bacteria. The lysis process was consistent with T7 phage. The size of the head was 80 nm × 110 nm, containing

**Table 1 Difference between T4 phage, T7 phage, λ phage and M13 phage.**

| Phage system | Genome | Characteristics | Life cycle | Splitting with other cellular proteins |
|---|---|---|---|---|
| T4 | Double-stranded DNA molecules | Cleavage phage | Long | Difficult |
| T7 | Double-stranded DNA molecules | Cleavage phage | Short | Difficult |
| λ | Double-stranded DNA molecules | Non-cleavable phage | Long | Difficult |
| M13 | Single-stranded DNA molecules | Non-cleavable phage | Short | Simple |

**Note:**
T4, T7, λ phage genomes are double-stranded DNA molecules; M13 phage genomes are single-stranded DNA molecules. T4 and T7 are characterized as cleavage phage; λ and M13 phage are non-cleavage phage. T4 and λ phages have long life cycles; T7 and M13 phages have short life cycles. T4, T7 and λ phage are difficult to separate from other cell proteins; M13 phage is easy to separate from other cell proteins.

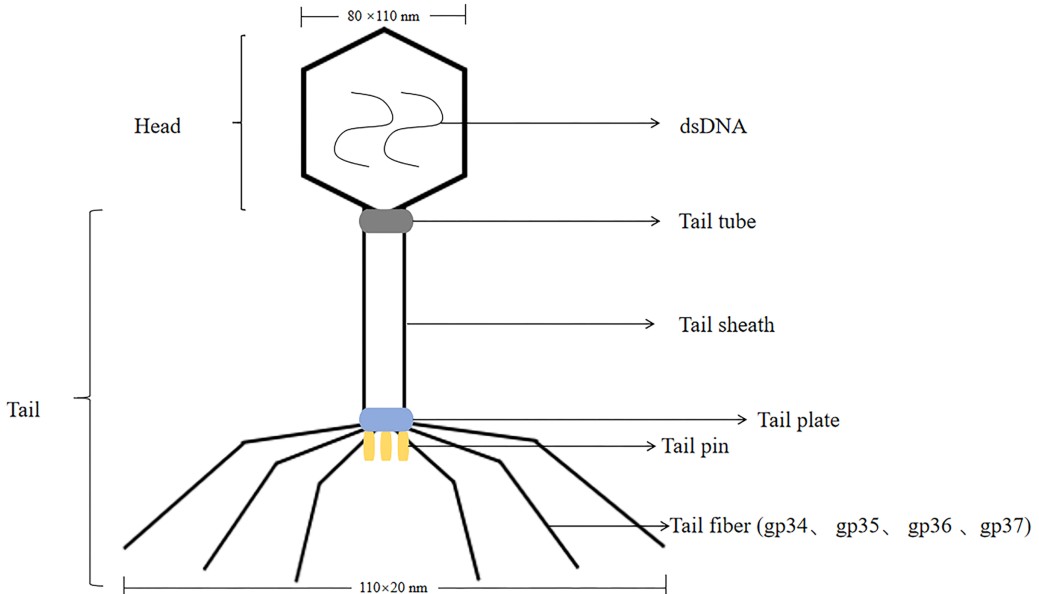

**Figure 2 The structure of the T4 phage.** T4 phages are double-stranded DNA. The size of the head is 80 nm × 110 nm. The size of the tail is 110 nm × 20 nm. The tail includes tail tube, tail sheath, tail plate, tail pin and tail fiber. The tail fibers consist of four different proteins are gp34, gp35, gp36, gp37.

double-stranded, thread-like DNA molecules with a relative molecular mass of $1.12 \times 10^8$ (*Hyman & van Raaij, 2018*). T4 phages had tails, double-stranded DNA viruses with a full cellular life cycle (*Gamkrelidze & Dąbrowska, 2014*). The tails of T4 phage consist of four different proteins were gp34, gp35, gp36, gp37. Isolation of T4 phage from phage by affinity chromatography can be achieved by using specific affinity tag, like glutathione tag (GST) or His tag (*Kutter et al., 1995*). The tail was composed of five parts: tail sheath, tail tube, tail plate, tail pin and tail fiber, with a length of 110 nm × 20 nm, and the tail filament can be extended with an amplitude of 140 nm (*Yap & Rossmann, 2014*; *Rao et al., 2023*; *Kuhn & Thomas, 2022*) (the structure of the T4 phage was shown in Fig. 2). T4 phage DNA ligase, originally from *Escherichia coli* cells infested with T4 phage, T4 DNA ligase can join both sticky ends and flush ends.

T4 phage acted in cancers, and mouse veGFR2 was constructed on the surface of T4 phage nanoparticles as a recombinant vaccine. T4-mouse veGFR2 recombinant vaccine

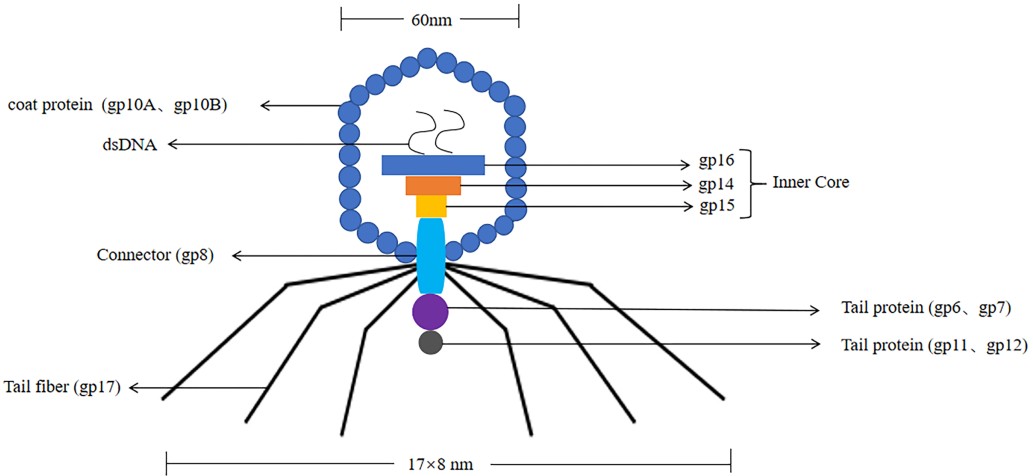

**Figure 3 The structure of the T7 phage.** T7 phages are double-stranded DNA. The size of the head is 60 nm. The size of the tail is 17 nm × 8 nm. Its coat protein are gp10A and gp10B.The gp14, gp15, gp16 are inner core in the head. Its connector is gp8. The tail proteins include gp6, gp7, gp11, and gp12. The tail fiber is gp17.                                    

was used as an antitumor drug (*Ren et al., 2011*). In addition, T4 phage was used to construct an anti-HIV vaccine, and immunization of mice with T4 phage carrying HIV antigens elicited strong cellular and humoral immunity. Effective demonstration of HIV antigens on phage T4 capsids provided a basis for the design of new HIV vaccines (*Sathaliyawala et al., 2006*).

## T7 phage

T7 phages were defined as one of seven phages that replicate in *Escherichia coli*, contain double-stranded DNA molecules, and were highly stable (*Deng et al., 2018*). The T7 phage genome was about 40 kb in size. The head had an icosahedral structure with a diameter of about 60 nm, and three structural proteins (gp14, gp15, gp16) were located in the head (*Matsuo & Fujisawa, 1972*). Its coat protein usually had two forms, namely 10A (344 amino acid residues) and 10B (397 amino acid residues), and the 10B coat protein region was present on the surface of the phage, so it was used to construct phage display systems (*Cerritelli et al., 2003*). Its connector was gp8. The tail was 17 nm × 8 nm in size, and the tail proteins included gp6, gp7, gp11, and gp12. The tail filament consisted of gp17 with six short tail filaments (The structure of the T7 phage was shown in Fig. 3). Therefore, it was widely used to screen proteins with different molecular weights and different affinities.

T7 phage was a cleavable phage. T7 phages generally infested bacteria in five steps. First, the tail filament protein of the phage bound to the receptor protein on the bacteria, this process was called the adsorption process (*Siegel & Summers, 1970*). The lysozyme was used to create an opening in the peptidoglycan layer of the bacterial cell wall to transfer the DNA of the phage into the bacteria, this process was called the infiltration process (*Krumpe & Mori, 2014*). The phage DNA was synthesized in the bacterium using the chemical components of the bacterium to synthesize the phage's own DNA or protein, this process was called synthesis (*Kulczyk & Richardson, 2016*). The newly synthesized DNA or

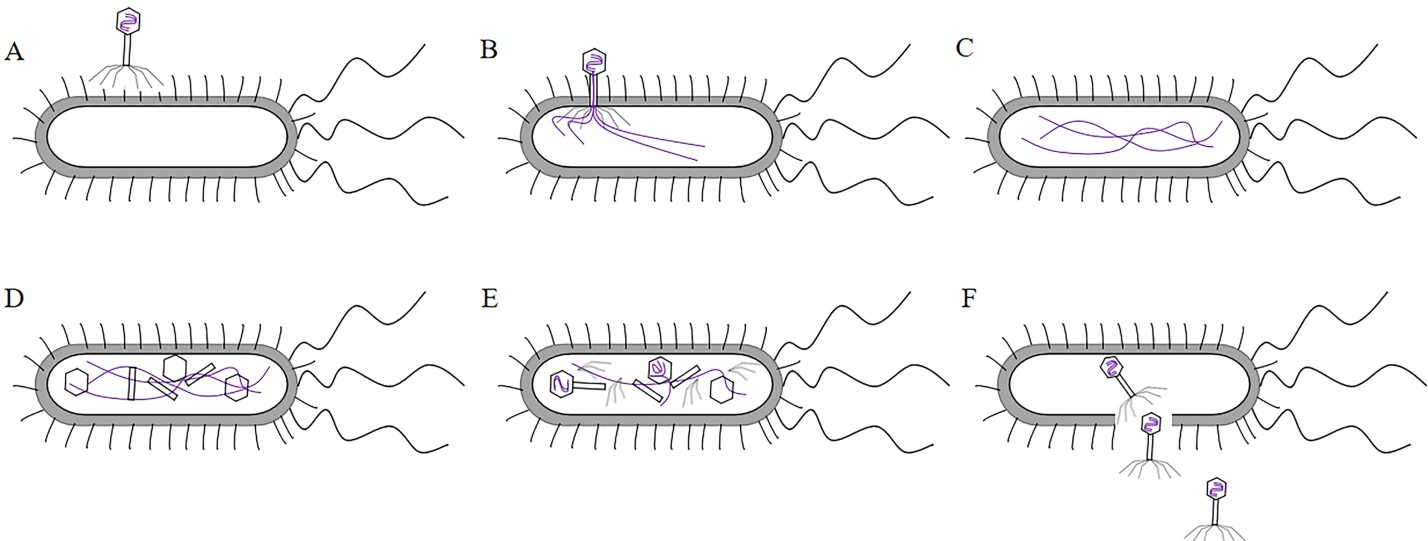

**Figure 4 The process of T7 phage infect bacteria.** (A)The tail filament proteins of the phage bind to receptor proteins on the bacteria, this process is called adsorption. (B) The phage invades its DNA into bacteria, this process is called infiltration. (C) The phage DNA is replicated in bacteria and is synthesized at an early stage. (D) The head, tail, and fiber proteins of the phage are made. (E) The phage DNA is assembled in the head, and assembled into zygote that is identical to the parental phage, this process is called assembly. (F) The bacteria are destroyed and the zygotic phage is released, this process is called release.

protein was replicated, amplified, and assembled into daughter phages that were identical to the parental generation, this process was called the assembly process (*Shin, Jardine & Noireaux, 2012*). Finally, these zygotic phages were released due to the rupture of the bacteria, this process called release (*Hernandez & Richardson, 2017*) (the T7 phage infected bacteria as shown in Fig. 4). Under optimal conditions, the T7 phage can complete the lysis process within 25 min, resulting in the death of *Escherichia coli*. At the time of lysis, it can produce more than 100 zygotes.

## λ phage

λ phage was a non-cleavable phage. The λ phage genome integrated into the host bacterial chromosome and did not produce zygote phages and did not cause bacterial lysis. However, λ phage DNA replicated like the bacterial genome (the λ phage infected bacteria as shown in Fig. 5). It was the earliest phage system and covered major capsid protein gpE and gpD (*Nicastro, Sheldon & Slavcev, 2014*). The λ phage was a mild phage of the long-tailed phage family with an icosahedral head of 55 nm in diameter and a long, thin tail filament at the end. The genome was a 48.5 kb linear double-stranded DNA molecule with sticky ends for single-stranded extension of 12 nucleotides, and the linear genome can be cyclized immediately after infection (*Casjens & Hendrix, 2015*). The λ phage assembled inside the host cell without the need to secrete foreign peptides or proteins in the bacterial cytoplasm. It can exhibit active macromolecular proteins (above 100 kDa) and proteins that were toxic to the host cell, and had an extremely wide range of applications (*Häuser et al., 2012*).

The defects of λ phage were that the genome was too large. There were too many enzymatic sites, it had 5 *Bam*H I sites (G↓GATCC), 6 *Bg* I sites (A↓GATCT), 5 *Eco*R I sites
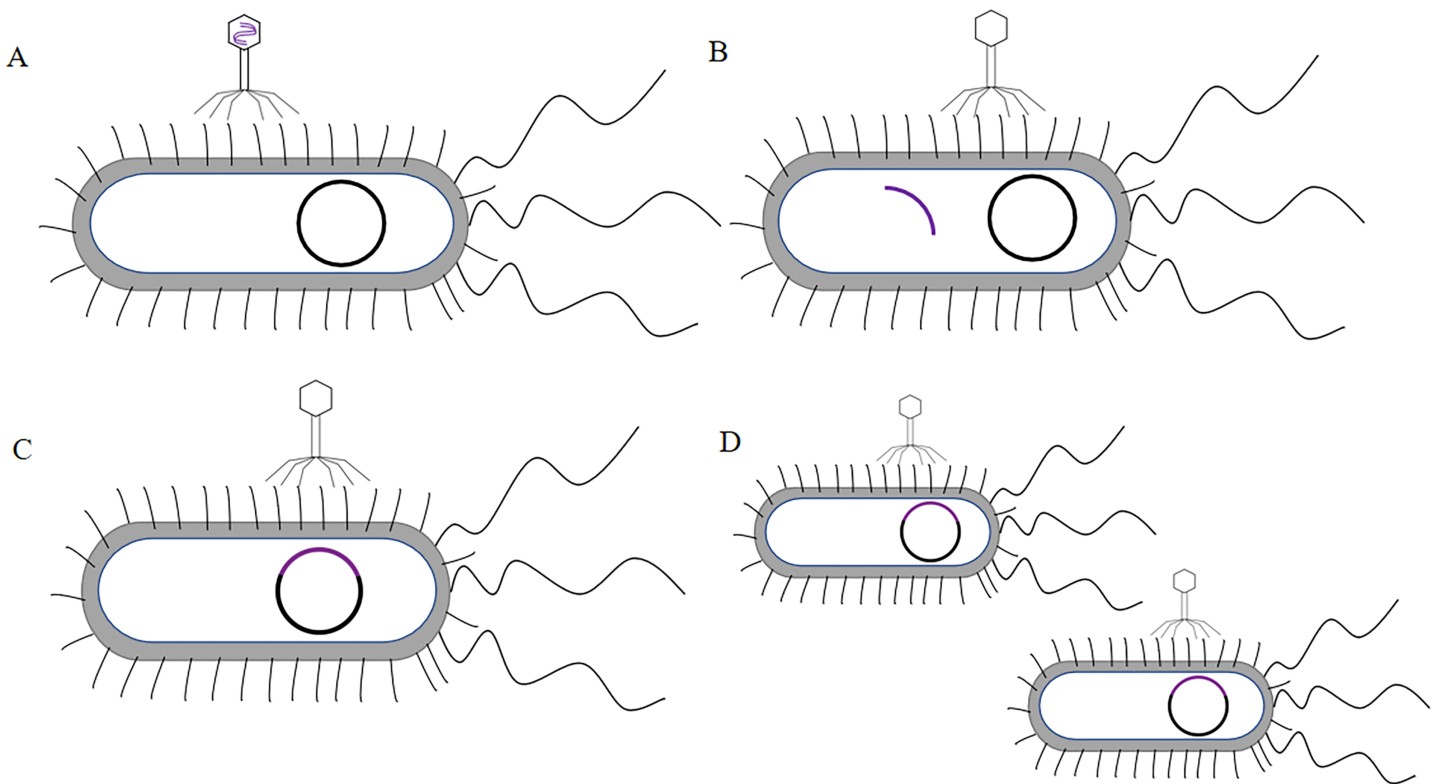

**Figure 5 The process of λ phage infect bacteria.** (A) The tail of the phage adsorbs to the surface of the bacteria, this process is called adsorption. (B) The phage DNA penetrates into the bacteria, this process is called penetration. (C) The phage DNA integrates with the bacterial chromosome, this process is called integration. (D) The phage DNA follows the replication of the bacterial genome.

(G↓AATTC). The recombinant λ DNA molecules were difficult to be directly introduced into the host cells, using λ DNA molecules packaged into viral particles *in vitro*, and then infected the host cells by the method of injection (*Nurmemmedov et al., 2012*; *Skalka, 1977*; *Sneppen, 2017*).

## M13 phage

The M13 phage has been confirmed as one of the most common filamentous phages, it involved the genetic information required for DNA replication and phage proliferation (*Parmley & Smith, 1988*). M13 phage was a non-cleavable phage. M13 phage was a single-stranded circular DNA virus with a 6.4 kb genome encoding 11 proteins, five of which were structural proteins, including the major coat protein PVIII and the minor coat proteins pIII, pVI, PVII and PIX (*Georgieva & Konthur, 2011*). The coat proteins most commonly used for phage display were pIII and pVIII and constructed pIII and PVIII display systems (*Smith, 1985*) (the structure of the M13 phage was shown in Fig. 6). M13 phage was by far the most widely used system for phage display technology (*Crameri & Suter, 1993*).

Most antibody phage display technology used M13 phages. The advantage of M13 phages as an antibody library display platform was that they were highly stable and can withstand high temperatures, prolonged storage, drying, acidic conditions, or disinfectant
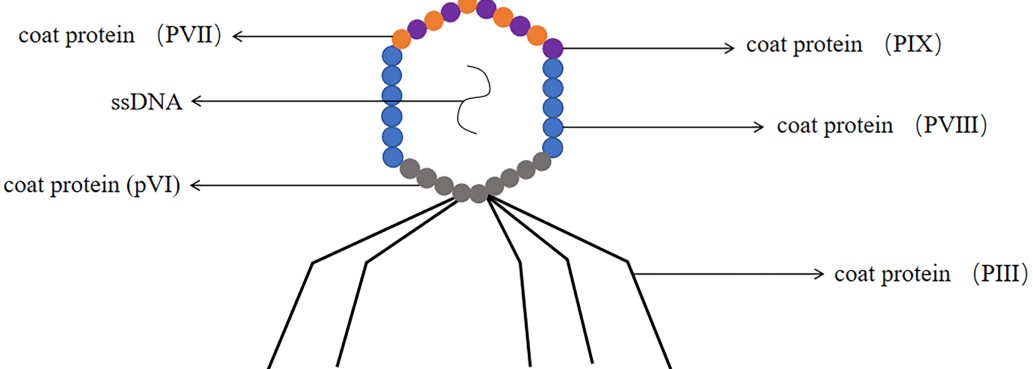

**Figure 6 The structure of the M13 phage.** M13 phage is single-stranded DNA. The major coat protein is PVIII and the minor coat proteins are pIII, pVI, PVII and PIX.

treatment (*Branston et al., 2013*). M13 phage minor coat protein pIII consisted of 406 amino acids and was distributed at one end of the phage particle (*Sidhu, 2001*). Generally a phage had 3–5 copies of pIII protein, which can insert foreign protein or peptide in the flexible linker region at the N terminus (*Rakonjac et al., 2017*). The main advantage of the pIII system was that there was no strict requirement for the size of the foreign protein to be displayed, and the system can be used to display proteins of larger molecular weight (*Allen et al., 2022*). The major coat protein PVIII had a molecular weight of 5.2 kDa and was mainly distributed on both sides of the phage particle, containing approximately 2,700 copies per phage (*Stopar et al., 2003*). Due to the high copy number of PVIII protein, this system was more suitable for screening low affinity ligands (*Adda et al., 2002*). Due to the small molecular weight of the protein, it was only suitable for displaying short foreign peptides. The foreign peptide was too large to influence the virus packaging and can not form a functional phage (*Stassen et al., 1995*).

The phage PIII minor coat protein consists of an N-terminal domain and a C-terminal domain. The N-terminal domain included N1 and N2, the C-terminal domain was involved in coat assembly (*Chang et al., 2023*). ScFvs fused to the N-terminal end of the M13 phage minor coat protein PIII (*Dunn, 1996*) and cloned the antibody fragment gene in large numbers to generate phage display single chain antibody libraries (*Ledsgaard et al., 2018*). Other genes were required to be provided by the helper phage, which can superinfect the phage-infected bacteria. The helper phage $M_{13}K_{07}$ was an engineered variant of the M13 phage that had an antibiotic resistance gene (usually to kanamycin) and a defective phage packaging signal, thus allowing preferential packaging of the phage vector to the helper phage DNA. Antibody phage display technology was rapidly replacing the classical hybridoma technique in many diagnostic and therapeutic applications. The advantage of phage display technology was that it allowed access to human antibody fragments without requiring preparation of mouse anti-humanized antibodies (*Pini & Bracci, 2000*). Using phage display methods required less time to generate antibodies than traditional hybridoma methods. Phage display antibody technology can be applied to construct human antibody with therapeutic value (*Ohara et al., 2006*; *Holt et al., 2000*).

Despite the introduction of various technological platforms, phage display technology was still the first choice for antibody generation because of the stability of the method and the ability to screen for the desired antibodies (*Harada et al., 2018*).

## Phage display single chain antibody libraries

Phage display libraries included natural antibody libraries, semi-synthetic antibody libraries, and synthetic antibody libraries. Natural antibody libraries were genetically derived from B-lymphocytes in blood, bone marrow, spleen and tonsils in humans or animals. The advantages of natural antibody libraries included the availability of human antibodies, targeting of all natural antigens, large enough libraries for direct access to high-affinity antibodies. But also it had more disadvantages, such as less library capacity than semi-synthetic and synthetic antibody libraries, time-consuming and laborious to build (*Lennard, 2002*; *Mahdavi et al., 2022*). Semi-synthetic antibody libraries, which were mainly based on the framework region of natural antibody libraries. The complementarity determining region 3 (CDR3) can be randomized and the length diversity can be changed to increase the capacity of the library, and some designers had randomized all six CDR regions to achieve a diversity of up to $10^8$ (*Zhou et al., 2011*). Synthetic antibody libraries, which were purely synthetic antibody libraries based on antibody gene libraries information. Fully synthetic antibody libraries required in-depth knowledge of the CDR regions of the antibodies, retaining the common or backbone part of the CDR regions, and designing replaceable gene regions to achieve a high degree of randomization. This created a huge library capacity (*Adams & Sidhu, 2014*). Antibody libraries with large capacity and rich diversity were easier to screen for ideal antibodies, so the quality of antibody libraries was generally evaluated by the two parameters of capacity and diversity (*Shim, 2015*). The capacity of fully synthetic antibody libraries was theoretically the largest, but there were key issues that need to be addressed, such as conserved sequencing of backbone regions (*Benhar, 2007*).

The phage display single chain antibody libraries were the recombinant phage libraries consisting of a large number of individual phages with different scFvs. In this large scFv libraries, after 3–4 rounds of screening, one or more scFvs will bind specifically to some molecule or the receptor on the surface of the target. This scFv provided an effective method for subsequent disease treatment or diagnosis (*Li et al., 2022*). The advantage of phage display single chain antibody libraries was that scFvs retained the binding specificity of the parental antibody, but also had better tissue penetration, which facilitated radiotherapy and diagnostic applications (*Rodríguez-Nava et al., 2023*; *Hosseinzadeh, Mohammadi & Nejatollahi, 2017*). For example, these fragments can penetrate tumors faster than intact antibodies. In radiation therapy applications, when scFvs bind to radionuclides, their clearance from the bloodstream was increased, thus minimizing exposure of healthy tissue (*Chester et al., 2000*). Compared to full-length mAb, scFvs lack the fragment crytallizable (Fc) region and were better therapeutic agents for many applications. ScFvs can be cloned and expressed in bacterial and mammalian cells, allowing for easy and cost-effective mass production (*Accardi & Di Bonito, 2010*). ScFvs had played an important role in the evolution of human life sciences. They were used as

tools to study protein function, cancer therapy, can bind to quantum dots and nanoparticles due to their small size. They can enter the body to localize targets, and serve as carriers for delivering drugs as well as nanoparticles (*Verhaar, Woodham & Ploegh, 2021*).

## Screening of phage display single chain antibody libraries

The results of phage libraries screening were affected by many factors. Firstly, the type of antibody library, affinity and screening methods. Secondly, it was influenced by environmental factors (*e.g.*, pH, temperature changes, time and other conditions), and again the antigen epitopes screened using phage libraries should be accurate and the purity of the antigen should be high (*Derda et al., 2011*). Screening methods included solid phase screening and liquid phase screening. Phage libraries were generally screened using solid phase screening, which involved a series of steps of "adsorption-elution-amplification" for biological elution, and several rounds of elution before the specific antibody was finally screened. The method of adsorption was to fix the target to a solid support (*e.g.*, a commonly used enzyme labeling plate) (*Jara-Acevedo et al., 2016*). The phage libraries were specifically bound to the target, and the non-specific phage was removed by washing, extreme pH, denaturant, ionic strength, limited proteolysis, ultrasound. Then the specific phage was eluted. After four or six rounds of elution, the eluted phages infected with *Escherichia coli* and amplified. The phage from the last round of elution were screened, and the selected phage were obtained through DNA sequencing (Screening process of phage display single chain antibody libraries in Fig. 7) (*Almagro et al., 2019*). During the biopurification process, it was necessary to screen phage clones with the highest affinity for the target by increasing the number and time of washes (*Pini & Bracci, 2000*). In contrast, the liquid phase screening method employed magnetic bead sorting, where the antigen was specially labelled and then incubated with phage libraries to ultimately screen for specific antibodies (*Aghebati-Maleki et al., 2016*). Compared with the solid phase screening method, the liquid phase screening method used a smaller amount of antigen (*Coomber, 2002*). When antigen was not available, cellular screening can be adopted, whereas cellular screening was difficult and tended to obtain considerable non-specific clones (*Hoogenboom et al., 1999*).

## The application of phage libraries

Over the last two decades, many methods have been used to screen proteins, among which surface display technology were very important. To be specific, phage display was the most convenient method, besides bacterial and ribosomal screening methods (*Goracci, Pignochino & Marchiò, 2020*). Accordingly, the application of the method will increase in the future (*Zambrano-Mila, Blacio & Vispo, 2020*).

Existing studies suggested that phage display technology will be combined with next generation sequencing to improve screening accuracy, reducing the occurrence of non-specific false positive clones. And it can speed up the identification process of target specific ligands (*Lemire, Yehl & Lu, 2018*). The use of phage display techniques to construct single chain antibody libraries for screening scFvs was elucidated in this review,

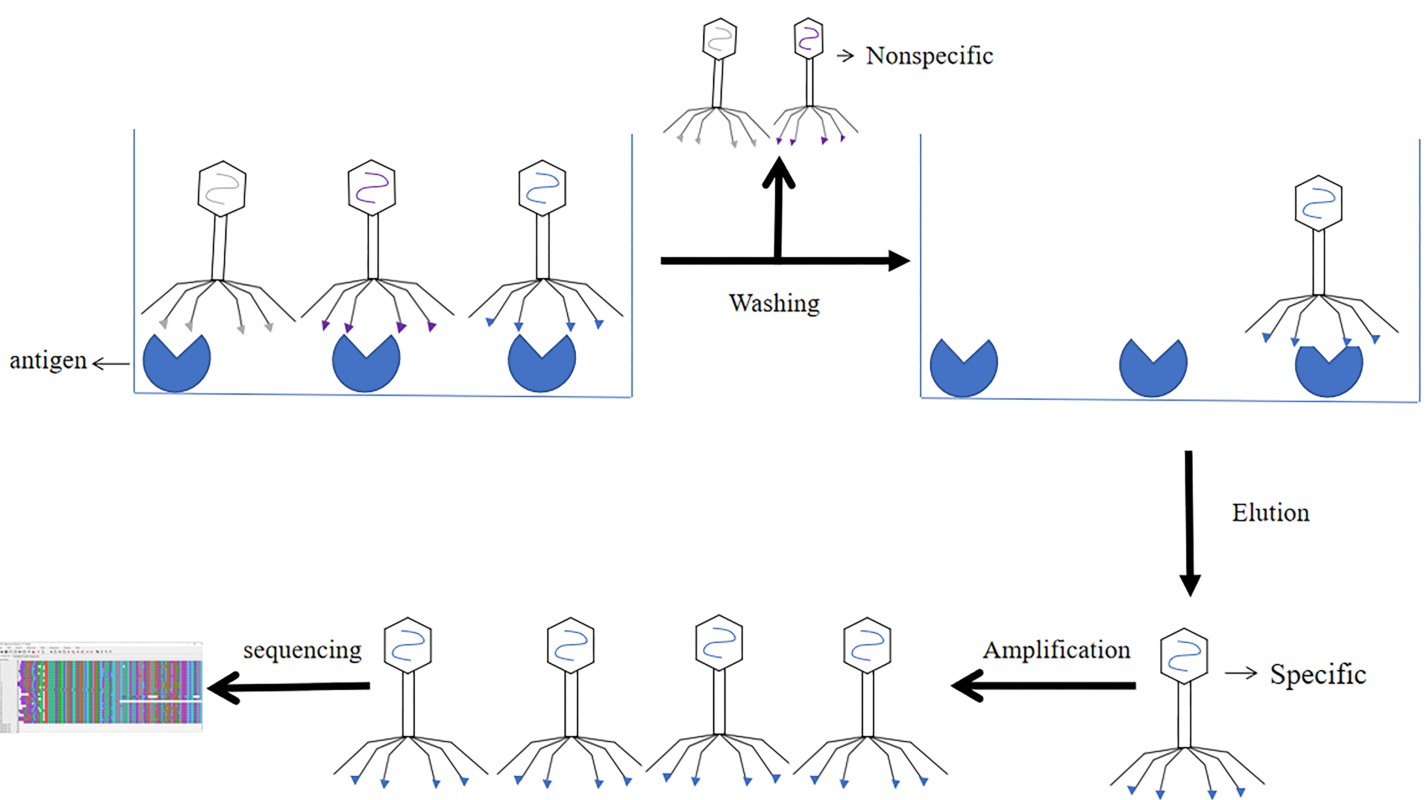

**Figure 7 Screening process of phage single chain antibody libraries.** The phage single chain antibody libraries are incubated with antigens, and the non-specifically bound phage is washed out, leaving the specifically bound phage for amplification and gene sequencing.

and it had played an integral role in vaccine development and cancer treatment (*Sokullu, Gauthier & Coulombe, 2021*). Cancer kills around 10 million people worldwide each year (*Torre et al., 2016*). The main problems are the lack of methods to target drugs to tumor sites and the lack of therapeutic agents with high specificity (*Lin et al., 2021*). In contrast, phage antibodies are diverse tumor binding libraries that may be useful for potential therapeutic agents and the development of new diagnostics (*Lindberg et al., 2021*). The development of a human lung diagnostic protein microarray with novel screening markers using the T7 phage cDNA library has been reported previously. In contrast to the conventional identification of the presence of lung cancer by lung tumor markers, the novel phage library technology enabled patients to detect and treat cancer at an early stage (*Hirsch et al., 2017*). Previous research has demonstrated that phage display technology has been employed in the treatment of cancer and the treatment of infectious diseases (*e.g.*, leprosy and infantile leishmaniasis) (*Alban et al., 2014*; *Coelho et al., 2015*; *Alban et al., 2014*). Combining cell surface-specific markers of parasitic infections with phage antibody libraries ultimately screens peptide compounds for the development of anti-parasitic drugs (*Kuzmicheva & Belyavskaya, 2016*). Furthermore, *Henry & Debarbieux (2012)* suggested that phages will reduce the production of pathogens in food, effectively reducing losses in the food industry. Now, many phage display antibodies have entered the biopharmaceutical market. Phage display technology played an important role in drug

discovery and development, and many phage molecules have been approved by the Food and Drug Administration, and many phage molecules have entered clinical trials. We can expect even greater improvements in the near future, and phage display will bring more contributions to oncology.

An ideal scFv library should produce a diverse range of scFvs with high specificity and high affinity. The size of the scFv library capacity determine the ability to screen for scFvs that bind to specific antigens. In general, the larger the library capacity, the higher the likelihood of screening for affinity scFvs will be. Several factors determine scFv library capacity (*e.g.*, the number of scFv genes, the efficiency of ligation, the efficiency of transformation into host bacteria for amplification and the efficiency of using phage infestation). The proportion of open reading frames is increased by reducing base mutations, frame displacement, and so forth to establish a big capacity antibody library (*Ling, 2003*). Currently phage display technology is one of the most convenient and most general used methods. This method exhibits a short experimental cycle and achieve favorable infiltration results. The capacity of the phage-infested antibody library is approximately twice that of the original antibody library. In general, the capacity of theoretical antibody library is about $1 \times 10^{7-8}$ pfu/mL, whereas the phage-infested antibody library can reach $1 \times 10^{11-13}$ pfu/mL (*Wang et al., 2021*). Indeed, there are some shortcomings in phage display single chain antibody libraries (*e.g.*, the low efficiency of phage infestation, the large number of false positive clones, and the high duplication rate of scFv genes in scFv libraries) (*Willats, 2002*).

As protein engineering and biotechnology have been leaping forward, the technology of phage display single chain antibody libraries establishment and its development in medical diagnostics and biosciences will advance, thus underpinning the promotion of biological and medical research (*Nagano & Tsutsumi, 2021*). In addition, the establishment of phage display single chain antibody libraries has become irreplaceable in the role of targeted cancer therapy (*Iezzi et al., 2018*). Phage library takes on a great significance in immunology, oncology, molecular biology, genetics, and pharmacology, and it has great possibility for the progress of discovering novel drugs in the future, thus making it a vital hotspot in biological research.

### Funding
This review was supported by the Key Research and Development Project of Heilongjiang Province of China (GZ20210101), the Cultivation Project of Heilongjiang Bayi Agricultural University (XDB-2016-22), and the Postdoctoral Scientific Research Start-up Fund of Heilongjiang (LBH-Q21158). The funders had no role in study design, data collection and analysis, decision to publish, or preparation of the manuscript.

### Grant Disclosures
The following grant information was disclosed by the authors:
Key Research and Development Project of Heilongjiang Province of China: GZ20210101.

Cultivation Project of Heilongjiang Bayi Agricultural University: XDB-2016-22.
Postdoctoral Scientific Research Start-up Fund of Heilongjiang: LBH-Q21158.

## Competing Interests

The authors declare that they have no competing interests.

## Author Contributions

- Xiaohui Zheng conceived and designed the experiments, performed the experiments, analyzed the data, prepared figures and/or tables, and approved the final draft.
- Qi Liu conceived and designed the experiments, performed the experiments, analyzed the data, prepared figures and/or tables, and approved the final draft.
- Yimin Liang conceived and designed the experiments, performed the experiments, analyzed the data, prepared figures and/or tables, and approved the final draft.
- Wenzhi Feng conceived and designed the experiments, performed the experiments, analyzed the data, prepared figures and/or tables, and approved the final draft.
- Honghao Yu conceived and designed the experiments, performed the experiments, prepared figures and/or tables, and approved the final draft.
- Chunyu Tong conceived and designed the experiments, authored or reviewed drafts of the article, and approved the final draft.
- Bocui Song conceived and designed the experiments, authored or reviewed drafts of the article, and approved the final draft.

## Data Availability

This is a literature review.

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
