# Peer review of "Advancement in the development of single chain antibodies using phage display technology"

_PeerJ, doi:10.7717/peerj.17143_

## Round 0.1 · original submission · Major Revisions

Please address the concerns of all reviewers and amend the manuscript accordingly.

**Language Note:** The review process has identified that the English language must be improved. PeerJ can provide language editing services - please contact us at [email protected] for pricing (be sure to provide your manuscript number and title). Alternatively, you should make your own arrangements to improve the language quality and provide details in your response letter. – PeerJ Staff

Reviewer 1 ·

Basic reporting

The introduction provides a comprehensive overview of monoclonal antibodies, genetically engineered antibodies, and the role of phage display technology in advancing antibody development. However, here are some comments and suggestions to enhance this section:

The introduction lacks a clear structure. Try to break it down into subsections that focus on specific topics like the historical development of antibodies, limitations, and the development of engineered antibodies.

The introduction needs a clear transition into what the review will cover specifically regarding phage display technology and its application in generating single-chain antibodies. Please consider concluding the introduction with a statement that directly introduces the scope of this review.

Experimental design

For the survey methodology, while the methodology briefly mentions downloading relevant literature and reading it, it's important to document the search process thoroughly. Keeping records of the search strategy, for example, the number of articles found, reasons for inclusion/exclusion, publication date and etc.

This review text should make a good balance between the technical details (about phage structures, genetic makeup, and their functions) and the explanations or examples. Some parts, especially those discussing the differences between various phage types, could provide more explanations.

Validity of the findings

The review covers various applications of phage display technology, such as cancer treatment and vaccine development. It is important to emphasize the impact of these applications by including specific case studies to illustrate their significance.

Reviewer 2 ·

Basic reporting

There are a lot of issues need to be considered or revised for a quality manuscript. Issues like format, professional writing, figures are detected, some examples are list below. Remember, below are just example, the author should also check other part of the paper, and make it professional, easy understanding, not misleading before submit the manuscript.
Line 24, scFv usually refer to single chain fragment variable in a lot of papers and research articles.
Line 36, "can treat multiple ....", better to use "can be used for..."
Line 49, "critical significance" only significance is enough
Line 53, Need to rephrase, there is no subject in the late half sentence
Line 59, formatting issue "and no"
Line 115-116, "Download the relevant literature, read it, summarize it, and write down my own ideas and thoughts" does not sound like professional in paper...
Line 144-145, This is not a complete sentence. can rephrase like: isolation of XXX can be achieved by using specific affinity tag, like XXX
Line 195-199, space issue, this paragraph has a larger line space than others
Line 207-213, same comments as above
Line 238, "antibody libraries and the The CDR3 region", and the The
Line 238, this is first time using CDR abbrevation, the author should have a full name writing of this term
Line 280-284, the whole four lines is just one sentence, too long, wordy, and can be misleading. Please rephrase the sentence.
Line 304, The use of phage display XXXX is elucidated XXX. In this sentence, the subject is "the use", so should use "is" instead of "are"
Line 306, "the main problems ..." "The main problems..."
Line 323-324, "Several factors are capable of determining" change to "Several factors determine"
Line 328, "Currently the use of phage display technique" change to "Currently phage display technique..."
Line 343, "progress of novel drugs in the future," change to "progress of discovering novel drugs in the future,"
Figure 1, hard to read the vector information...

Experimental design

1, The author first introduced the background of phage display, but the some information is misleading. for example, line 76-103, the author mentioned "the phage vector pcantab5e" and helper "M13K07". according to words by the authors, it looks like there is only one vector pcantab5e system for phage display, but there are more options like pComb3 Phage vector, if use phage vector of pComb3, then helper phage could be VCSM13. the whole sentence is misleading. The author should rephrase sentence like "Subsequently, the linked scFv is ligated to the phage vector like pcantab5e ..."
2, Survey methodology
there is a big issue here, for example line 113, "we searched the keywords "phage display, single chain
114 variable antibody fragments, phage libraries " in PubMed. Is this key word one example for searching ? all just the only one being used? if if it later, the search is biased and not comprehensive. Sometime, searching same topic can use different phrase of key work, for example, using "single chain fragment variable" instead of "single chain variable antibody fragments" may result in a different set result.
3, line 140-150, and also other part of paper, when author describe the phage, it would be nice to have a general figure about phage shape, size, tail ...
4, as whole review paper, there are only two figures, which makes the manuscript very week, it would be nice if the author add more details about the key processs, the key objects

Validity of the findings

As a review paper, I feel like this manuscript is not in-depth enough. It is more like a general report or summary of previous paper or research. The phage display technology has been developed and implemented in industry for a long time, I do not find the big impact or novelty for this review. Especially, the author describe each aspects of the paper in a very general way.
I also noticed that, almost one end of sentence, there is one reference paper, for a comprehensive review paper, a lot of descriptions can related to several different reference, sometimes a single reference paper can appear several time in the whole paper. one example, line 314-315, "in the treatment of cancer and
the treatment of infectious diseases", the technology can be used for several different disease, then you should have more than one reference paper to prove this.

Overall, I think the manuscript need big edition, including professional and scientific writing, in-depth discussion of the topic.

Reviewer 3 ·

Basic reporting

The submitted review by Chunyu Tong et al. delves into the enhanced development of single-chain antibodies through phage display technology. The comprehensive discussion encompasses various antibody types, phage display systems, libraries, screening methods for antibody phage libraries, and applications. Additionally, the manuscript cites a recent review by Long Li et al. (2022) in Compr Rev Food Sci Food Saf., which focuses on single-chain fragment variables produced by phage display technology in the context of food display. The authors appropriately reference other research articles in their review. The English of the manuscript needs to be improved. While the review is of broad and cross-disciplinary interest, some refinement is suggested in the additional comments.

Experimental design

N/A

Validity of the findings

N/A

Additional comments

1. Please enhance the visibility of Figure 1 for readers by increasing the font size, and altering colors might also be beneficial.
2. I suggest considering a title rearrangement for the review, such as "Advancement in the development of single-chain antibodies using phage display technology."
3. Line 42, kindly include a sentence indicating the limitation of using monoclonal antibodies.
4. Line 54, provide more details in the sentence and cite the research article stating, "The development of cancer primarily involves over one signaling pathway due to the complexity of cancer cells."
5. Line 75, it would be valuable if the authors addressed the limitations of bispecific and chimeric antibodies.
6. Line 116, as there are seven authors in the manuscript, I recommend changing the word "my" to "our."
7. Line 117, including a figure with a schematic representation of the different phages discussed in the manuscript would aid readers.
8. Line 257, please include the original research article in the manuscript, along with the review by Long Li et al. (2022) in Compr Rev Food Sci Food Saf.
9. A couple of sentences need to be rewritten for a more scientific tone. For instance, on line 32, "Since 1975 Kohler… Hybridoma cells" could be revised as "Since 1975, Kohler[1] has generated specific monoclonal antibodies against antigens by fusing mouse bone marrow cells with mouse spleen cells, resulting in the production of hybridoma cells." Another example is on line 59, "Any specific antibody … moral issues" which could be rephrased as "Antibodies can be derived from animals; however, obtaining antibodies from humans poses ethical and moral challenges, preventing their acquisition."

---

## Round 0.2 · Minor Revisions

Please address the remaining concerns of the reviewers, fix the formatting issues pointed out by reviewer #2, and provide a clarification requested by reviewer #3

Reviewer 1 ·

Basic reporting

Thanks for the authors' responses, and the revision looks good to me.

Experimental design

no comment

Validity of the findings

no comment

Additional comments

no comment

Reviewer 2 ·

Basic reporting

All the issues mentioned in the last review were corrected.
Just some small formatting issues, for example, sometimes a period is inside a parentheses, sometimes outside. Sometimes there is an extra period before a parentheses. Also some parts, there is missing space before a new sentence. Formatting issues like those can be improved.

Experimental design

The survey methodology part has been improved. A more detailed explanation including figures has been added.

Validity of the findings

This part has been revised too.

Reviewer 3 ·

Basic reporting

The submitted review by Chunyu Tong et al. delves into the enhanced development of single-chain antibodies through phage display technology. After thorough revisions by the authors, the review has significantly improved. I recommend publication of this revised manuscript in PeerJ with a minor suggestion as follows:

Line 42-43, "The preparation process of monoclonal antibody was cumbersome, and the preparation technology was higher." Could you please provide more clarification on what you mean by 'preparation technology was higher'?

Experimental design

N/A

Validity of the findings

N/A

Additional comments

N/A

---

## Round 0.3 · accepted · Accept

All remaining concerns of the reviewers were adequately addressed, and the revised manuscript is acceptable now.